# CRISPR/Cas9 Based Cell-Type Specific Gene Knock-Out in *Arabidopsis* Roots

**DOI:** 10.3390/plants12122365

**Published:** 2023-06-19

**Authors:** Meng Li, Xufang Niu, Shuang Li, Shasha Fu, Qianfang Li, Meizhi Xu, Chunhua Wang, Shuang Wu

**Affiliations:** College of Life Sciences and Horticultural Plant Biology Metabolomics Center, Fujian Agriculture and Forestry University, Fuzhou 350002, Chinashuangli_lis@163.com (S.L.);

**Keywords:** CRISPR/Cas9, genes function, cell-type-specific gene knock-out, root

## Abstract

CRISPR/Cas9 (hereafter Cas9)-mediated gene knockout is one of the most important tools for studying gene function. However, many genes in plants play distinct roles in different cell types. Engineering the currently used Cas9 system to achieve cell-type-specific knockout of functional genes is useful for addressing the cell-specific functions of genes. Here we employed the cell-specific promoters of the *WUSCHEL RELATED HOMEOBOX 5 *(*WOX5*), *CYCLIND6;1* (*CYCD6;1*), and *ENDODERMIS7* (*EN7*) genes to drive the Cas9 element, allowing tissue-specific targeting of the genes of interest. We designed the reporters to verify the tissue-specific gene knockout in vivo. Our observation of the developmental phenotypes provides strong evidence for the involvement of *SCARECROW* (*SCR*) and *GIBBERELLIC ACID INSENSITIVE* (*GAI*) in the development of quiescent center (QC) and endodermal cells. This system overcomes the limitations of traditional plant mutagenesis techniques, which often result in embryonic lethality or pleiotropic phenotypes. By allowing cell-type-specific manipulation, this system has great potential to help us better understand the spatiotemporal functions of genes during plant development.

## 1. Introduction

Functional genomics studies in plants require the generation of loss-of-function mutations in specific genes of interest. The conventional strategy for mutagenesis in plants is to generate stable and heritable mutations, which often result in embryonic lethality or cause pleiotropic phenotypes. In addition, most plant tissues are composed of functionally and morphologically distinct cell types, making it challenging to dissect the functions of a particular gene by globally disrupting the gene in whole plants. Tissue-specific or cell-type-specific mutagenesis can overcome these limitations and provide unprecedented insight into the role of a gene in a specific cellular context. Previous attempts at tissue-specific RNAi have been made in plants, but small RNAs can be mobile, limiting the tissue specificity of gene silencing [1]. With the development of gene-editing technology, the clustered regularly interspaced short palindromic repeats (CRISPR)-associated protein 9 (Cas9) nuclease creates DNA double-strand breaks at specific genomic sequences, resulting in edited sequences [2,3,4]. Gene-editing tools mediated by the Cas9 system mainly include Cas9 endonuclease and guide RNA (gRNA) [5,6]. gRNA acts as a guide for Cas9 to complex with endonuclease, which cuts the DNA of target genes. Constitutive promoters with sustained high expression in all cell types have been widely used in transgenic systems. However, the constitutive expression of Cas9 often leads to problems, such as high off-target rates and prevents the dissection of gene function in a tissue-specific background. Much effort has, therefore, been devoted to modifying and expanding this system to make it more versatile. In many model systems, including zebrafish, drosophila, and mice, researchers have recently developed a tissue-specific knockout technique using the CRISPR-Cas9 system [7,8,9,10,11,12]. In plants, studies have aimed to generate stable genetic mutants by expressing the Cas9 system in egg or germ [13,14,15]. Recently, pioneering efforts in tissue-specific CRISPR and inducible gene editing in specific cell types in plants have also been reported [13,16,17]. Genes specifically expressed in the meristem or embryo are preferentially used in the modified Cas9 system, such as YAOZHE (YAO), INCURVATA (ICU), and cell division control protein 45 (CDC45) [18,19,20,21]. Although the specific knockout system does not significantly improve the editing efficiency, it has achieved the knockout of the gene of interest in the target tissue.

*Arabidopsis thaliana* has long been the model organism for the study of plant development. Owing to its stereotyped anatomy, the *Arabidopsis* root has long been used to study plant organogenesis. From the inside to the outside, *Arabidopsis* root is composed of concentric rings of different cell layers, including stele, endodermis, cortex, and epidermis [22,23,24]. These different cell types are all derived from a stem cell niche (SCN) located at the root tip, which serves as a source of new cells for root growth. Continuous root growth requires the maintenance of the SCN, which relies on the quiescent center cells (QCs) located in the center of the SCN to remain in a non-dividing and non-differentiating state [25,26,27]. The quiescent state of QC cells is essential for suppressing cell differentiation in neighboring stem cells. The regulation of the SCN requires the coordination of several key regulators, in which SCR has been reported to act in concert with SHR to maintain SCN homeostasis [28,29,30]. In addition, SCR/SHR are essential for the development of cortical and endodermal cells in plant roots. Cortex endodermis initials (CEIs), located on either side of the QCs, give rise to endodermis and cortex cells by periclinal division. The endodermis has special differentiation systems (Casparian strips and suberin) that are particularly important for plant roots to conserve water and to resist external stresses [16,31,32]. *SCR*, which is expressed in QCs, initial ground tissue cells, and endodermal cells, is involved in maintaining the SCN and regulating endodermis division. Previous studies on SCR function were based on phenotypic observations of *scr* mutants [28,29,30,33]. The pleiotropic effect of *SCR* mutations on many parts of the plant prevents a precise assessment of the role of SCR in the SCN and CEIs.

After CEI division, endodermal cells must undergo rapid cell expansion, which is regulated by the GAI protein of the DELLA family, a key repressor of the gibberellin (GA) signaling pathway. When GA is perceived by its receptors, DELLA proteins are induced to degrade, which switches on GA signal transduction. In the *gai* mutant (a non-GA-degradable mutant form of GAI), cell division in the root meristem is significantly reduced and cell elongation in the transition zone is also dramatically inhibited [34]. Although the endodermis-specific perturbation of GA signaling is sufficient to cause root phenotypes, *GAI* expression is not restricted to the endodermis [35]. Therefore, tissue- or cell-type-specific gene silencing is crucial for dissecting the function of such genes with a broad range of expression.

In recent years, plant scientists have increasingly focused on investigating the tissue- or cell-specific aspects of plant development. The Cas9 element has been successfully driven by gene promoters that are specifically expressed in differentiated tissue cells, such as *SOMBRERO* (*SMB*) and *hydroxycinnamoyl transferase* (*HCT*) [36,37,38,39,40]. Despite these proof-of-concept studies, it remains unclear whether cell-specific knockout can be widely achieved in many other cell types. This is particularly important for key factors that regulate tissue development in the early stages, when different cell types are just being initiated. It is, therefore, important to explore other promoters that are active in specific stem cells. To validate the tissue-specific function of SCR and GAI in maintaining SCN and endodermal cells, we modified the CRISPR-Cas9 system to drive the expression of the functional element Cas9 in specific cell types in this study. We used the promoters of *WOX5*, *CYCD6*, and *EN7*, which are specifically expressed in QC cells, CEIs, and endodermal cells, respectively, to drive Cas9. We successfully knocked out *SCR* or *GAI* specifically in QC cells, CEIs, and the endodermis, respectively. In addition, we employed both promoter-GUS reporters to allow the real-time visualization of Cas9 expression, and GFP fluorescent reporters of the corresponding genes to reflect the gene editing events in vivo. Using this system, we were able to dissect the distinct roles of SCR in QC cells and CEIs, and verify the essential role of GAI in the endodermis.

## 2. Results

### 2.1. Specific Knockout of SCR in QC Leads to Impaired QC Function and SCN Maintenance

With the exception of *WOX5*, many genes that have been reported to affect SCN activity are not specifically expressed in root stem cells (Figure 1A). In order to dissect the specific role of these genes in root stem cells, we need to generate cell-type-specific gene knockouts that only abolish the function of these genes in the SCN. We first chose SCR, a transcription factor expressed in the QCs, CEI, and endodermal cell lineages (Figure 1A). To specifically knock out *SCR* in the QCs, we constructed a Cas9 expression cassette under the control of the QC-specific promoter of WOX5. To ensure that Cas9 expression was restricted to the target cell, we added a GUS reporter and nuclear localization signal (NLS) to the expression cassette driven by the same promoter. Next, we combined gRNAs targeting SCR, driven by a U6 promotor with the *pWOX5:Cas9*-*pWOX5-GUS* expression cassette. This binary vector allowed the simultaneous expression of *SCR*-targeting gRNAs and Cas9 in QC cells. The specific expression driven by pWOX5 was confirmed by the QC-restricted GUS staining (Figure 2A–C). To verify the editing of *SCR*, we took advantage of the visible reporter of *pSCR:SCR-GFP*. In the control, SCR-GFP was detected in QC, CEI, and all endodermal cells (Figure 2D). In the *pWOX5:Cas9*; *pSCR:SCR-GFP* lines, we observed a clear reduction in SCR-GFP in QC cells, while the SCR-GFP in CEI and endodermis was maintained at the WT level. In some roots, SCR-GFP was substantially reduced in all QC cells, whereas in some other roots, SCR-GFP was reduced in only one of the QC cells, which may reflect the corresponding editing events in these cells (Figure 2E,F).

Consistent with the specific reduction of SCR-GFP in the QC, propidium iodide (PI)-stained roots of *pWOX5:Cas9-SCR* lines showed an irregular QC shape, as well as disorganized columella cells, a phenotype similar to the *scr-4* mutant, in which QC function is defective (Figure 3A–D) [28]. Further mPS-PI and Lugol’s staining verified the loss of repression of CSC differentiation in *pWOX5:Cas9-SCR* roots, suggesting that the normal QC function was impaired (Figure 3E–L). Although QC was affected in both *scr-4* mutant and *pWOX5:Cas9-SCR* lines, CEI division was only disrupted in *scr-4*. This result supports the conclusion that the specific *SCR* knockout allows us to dissect the role of SCR in QC cells.

### 2.2. Specific Knockout of SCR in CEI Allows Dissection of SCR Activity in Cell Division

To further test the efficacy of the tissue-specific Cas9 system, we attempted to knock out *SCR* specifically in CEI. To this end, we replaced the pWOX5 in the *pWOX5:Cas9-SCR* vector with the previously reported CYCD6;1 promoter [41]. It was shown that SHR and SCR promoted asymmetric division in CEI by activating a D-type cyclin (CYCD6;1). Using the previously published promoter region of CYCD6;1, we were able to specifically express Cas9 and GUS reporters in the CEI (Figure 1A and Figure 2A–C). In *pCYCD6;1:Cas9-SCR* roots, we observed a significant decrease in SCR-GFP in the CEI (Figure 2D). Interestingly, some early endodermal lineages often showed reduced SCR-GFP in the roots with *SCR* edited in the CEI, suggesting that these early endodermal cells may be derived from the edited CEI (Figure 2D). In addition, we were surprised to find that some QC cells also appeared to be edited in *pCYCD6;1:Cas9-SCR* roots (Figure 2E,F), suggesting that the CYCD6;1 promoter may promote weakly leaky expression in the QC. In all roots examined, we could not identify the CEI with the background fluorescence of SCR-GFP. Considering the transient expression pattern of *CYCD6;1* during the cell cycle in the CEI, it is possible that the CYCD6;1 promoter was not strong and durable enough to completely remove SCR. Nevertheless, we still observed irregular division patterns in these edited CEI cells, suggesting that reduced SCR levels may also affect the regulation of cell division in the CEI (Figure 3A–L).

### 2.3. Endodermis-Specific GAI Knockout Causes Stunted Root Phenotype

To test whether the CRISPR-Cas9 system could edit other loci and be functional in other cell types, we targeted GAI, a key repressor of the GA response [35]. It was previously reported that the specific expression of *gai*, a mutant form that disrupts the GA response, in the endodermis was sufficient to cause aberrant cell expansion in the cortex and other cell layers [34]. It was previously reported that the *EN7* gene is specifically expressed in the endodermal cell layer [41,42,43] (Figure 1A). In this study, we generated an endodermal *GAI*-targeting construct using the pEN7 promoter (Figure 1A). To ensure that the Cas9 was restricted to the endodermal cells, we added a GUS reporter and a nuclear localization signal (NLS) to the expression cassette driven by the EN7 promoter (Figure 1B). GUS staining showed clear endodermal-specific expression driven by the tested EN7 promoter (Figure 2A–C). In many transformants of *pEN7:Cas9-GAI* lines, root growth was significantly reduced (Figure 4A,B). In these inhibited *pEN7:Cas9-GAI* roots, we found a markedly reduced cortex length, a phenotype similar to the *gai* mutant (Figure 4C–G) [34]. In addition to the cortex length, the number of root meristem cells was also slightly reduced in the *pEN7:Cas9-GAI* roots (Figure 4C–G). Owing to the non-specific expression of *GAI*, we could not fully achieve the shortened cortex cell phenotype of *gai* mutants by knocking out the *GAI* gene in the endodermis (Figure 4D,E,G). Taken together, our results provide a proof-of-concept for the cell-type-specific knockout of functional genes in *Arabidopsis* roots. This approach can be further modified by combining different promoters and sgRNAs, or even inducible expression cassettes, to make the system more versatile. This technology represents a powerful method for the functional dissection of genes in specific cell types and also overcomes the limitations of lethality and pleiotropy caused by the conventional gene knockout approaches.

## 3. Discussion

Cell-type-specific gene knockouts are important in plant genetics because they allow researchers to study the function of specific genes in a more targeted and controlled manner. This is particularly useful because plants are complex organisms with a variety of different cell types that perform specific functions. Many key regulators of plant development have been shown to be versatile, with broad expression in many different cell types. In addition, almost all plant cells are connected by the plasmodesmata and communicate with each other during development. Changes in one cell type can affect the development and physiology of neighboring cells in plants. By being able to knock out genes in a specific cell type, researchers can gain a better understanding of the role that that specific gene plays in a restricted context. In addition, breeders have long been troubled by the fact that mutations in an important gene often cause other pleiotropic phenotypes in addition to the desired traits, which can interfere with the use of the important genes. Cell-type-specific gene knockout can also be used to develop new plant-breeding techniques that can help to create crops with improved traits in specific organs or tissues, without influencing other parts of the crop.

*Arabidopsis thaliana* has been widely used as a model system in which tissue patterning is stereotyped and simple. In recent years, several attempts have been made to develop tissue-specific knockout systems in *Arabidopsis*. Initial pilot results achieved *Arabidopsis* transgenic lines with 10% editing efficiency using specific promoters to drive Cas9 in the root cap, stomata, and lateral roots [13]. This elegant work provides a proof-of-concept, demonstrating the feasibility of knocking out genes in specific organs or cells. However, this pilot study only looked at genes with specific expression patterns or knocked-out genes in a broad context in the lateral root. Another recent study went further and knocked out *PLT1/2* specifically in the root epidermis using the WEREWOLF (WER) promoter, and successfully achieved a cell division phenotype in lateral root development [17]. An advantage of this system is that it includes the inducible system, which extends the capacity for the spatiotemporal control of gene knockout in specific cell types. However, according to a previous report, PLT1 and 2 are able to move between cells in *Arabidopsis* roots. Therefore, it needs to be further validated that the functional abolishment of PLT1/2 in specific cells is not affected by the potential cell-to-cell movement of the proteins. In this study, we chose *SCR* and *GAI*, both of which have been reported to function in different cell types and for which no cell-to-cell protein movement has been observed. This makes them suitable targets for testing whether we can dissect the gene function in a specific cell type from its broad expression range. However, we did not find a significant difference in the editing efficiency in stem cells and the endodermis (Table 1) compared with the previous study in other organs and tissues.

Although a role for specific knockout systems has been achieved in *Arabidopsis*, the successful application of the system in crop plants is of greater importance for improved breeding. However, most crop plants have more complex tissue anatomy. For example, *Arabidopsis* roots have only a single layer of cortex, whereas most crop plants form multiple layers of cortical cells. Although there is evidence that many of the identified regulators have stereotyped expression patterns in both *Arabidopsis* and crops, the efficiency of cell-specific promoters previously identified in *Arabidopsis* needs to be validated. Another bottleneck issue is the successful delivery of the gene-modification components into crops. Many improvements have been made in the genetic transformation of crops. The activation of *Wuschel 2* (*WUS2*) in sorghum can simultaneously improve the targeted editing efficiency of Cas9 and transgene efficiency, as well as shorten the transformation cycle [44]. With many such breakthroughs in gene transformation techniques, more attention should be paid to the comparative evaluation of spatio-temporal regulatory promoters between different plant species in the future. Recently, an interesting study on *Arabidopsis thaliana* and *Brassica rapa* has been reported, in which a gene-edited *Arabidopsis* root was grafted to the aboveground tissue of *Brassica rapa*, and the mobile gene-knockout components in the *Arabidopsis* root could generate gene editing events in the grafted *Brassica rapa* [45]. It would be interesting to explore the possibility of combining tissue-specific gene knockout and these reported grafting-based mobile editing systems. In addition, foreign DNA in plant cells sometimes occurs during biotic stresses, and the cell-specific gene knockout technique may provide a viable strategy to eliminate the effect of biotic stresses in certain cell types without affecting the unaffected cells.

When modifying the Cas9 system for tissue-specific knockout, promoter selection is a key factor to consider. Previous studies have shown that promoters influence the editing efficiency and heritability of Cas9 at targeted genes. Although we did not find significant differences in editing efficiency in different specific cells, the different expression levels of Cas9 under cell-specific promoters could affect the editing efficiency. Bioinformatic approaches need to be developed to identify the critical elements in different promoters that affect the expression levels. Another issue that needs to be addressed for the tissue-specific knockout system is the verification of the gene-editing event in specific cells. As gene editing is independent and stochastic in different cells, the separation of the edited cells is essential for further PCR verification. To date, the successful separation of the edited cells has been achieved in leaf cells. However, protoplasting small amounts of stem cells in *Arabidopsis* roots appears to be extremely difficult. In addition, we used GUS as a reporter tag in this study, which made it difficult to achieve cell separation and, therefore, limited the verification of gene editing in individual cells. Alternative tags suitable for fluorescence sorting techniques should be included in future improvements. To overcome this obstacle, we utilized a dual reporter system, employing both promoter-GUS reporters, which allowed the real-time visualization of Cas9 expression, and GFP fluorescent reporters of the corresponding genes, which reflected the gene-editing events employed in vivo. Using this dual reporter system, we clearly observed that SCR-GFP was significantly attenuated or disappeared in QC cells, whereas SCR-GFP was unaffected in other cell types. Thus, the cell-specific knockout technique, combined with visible reporters, can be used to determine whether a gene has been successfully knocked down or knocked out in individual cells.

In summary, the Cas9 system has revolutionized the field of genetics and plant biology in recent years. Precise and efficient gene editing in specific cells has opened up new avenues for studying specific developmental processes. In addition, the ability to knock out genes in specific cell types with the Cas9 system can significantly advance our understanding of the role that specific genes play in a particular genetic background and context. In addition, the Cas9 system with cell-specific expression can also be used to create transgenic organisms with desired traits in limited tissues, providing new opportunities for the development of improved crops.

## 4. Materials and Methods

### 4.1. Plant Materials and Growth Conditions

All *Arabidopsis* seedlings were grown under the condition of a 16 h light/8 h dark cycle at 22 °C. The *Arabidopsis thaliana* Columbia ecotype (Col-0) was used as the wild type for all observational experiments. gRNA-targeting genes *SCR* (AT3G54220) and *GAI* (AT1G14920) were designed by http://crispr.hzau.edu.cn/CRISPR2/ (accessed on 18 May 2023). The *scr-4* (CS6505) and *gai* (CA63) mutants were in the Col-0 (WT) background and provided by the TAIR website mutant repository. Homozygous T3 seeds of all lines after sterilization with chlorine were germinated after chlorination and vernalization for 2 days at 4 °C in the dark. All plants were cultivated vertically on 1/2 MS medium, which contained 1% sugar and 1% sucrose.

### 4.2. Plasmid Construction and Plant Transformation

We used the restriction enzymes BbsI and AscI to cut pGWB604 (Appendix A), and then used the enzyme-digested product as the template for PCR with primers BbsI-F and AscI-R, or AscI-F and BbsI-R. The PCR products were then combined to form the expression vector. Next, the R4-CmR-ccdB-L1 fragment was amplified using R4L1-F and R4L1-R as primers and plasmid pGWB632 as a template (Appendix A). This fragment was then connected to the linearized pAtU6-sgRNA-Cas9 plasmid (Appendix A), which was cut by the restriction enzymes XmaI and NcoI, through homologous recombination. This resulted in the vector skeleton of pAtU6-sgRNA-R4-CmR-ccdB-L1-Cas9 being obtained. The amplified fragment was then amplified using pAtU6-F and pCas9-R as primers, and the final expression vector, 632-pAtU6-sgRNA-R4-L1-Cas9-R4-R2-eGFP-Tnos, was then obtained after a final binding with the HindIII restriction enzyme. The resulting binary vectors were introduced into the *Agrobacterium strain GV3101-pMP* and transformed into the *Arabidopsis thaliana* ecotype Col-0 using the standard floral-dip method. Transgenic plants were screened for resistance to Kanamycin (Kana) in soil. Three independently transformed lines were analyzed and the homozygous T3 seeds of them were chosen for further analysis. All primer sequences are listed in Appendix A.

### 4.3. Staining

The β-Glucoronidase (GUS) staining solution was prepared as previously described [46]. The *Arabidopsis* seedlings were incubated in the GUS solution (0.5 mg/L) for 2 h at 37 °C. Propidium iodide (PI) staining was used to mark the cell wall and observe the cell morphology; the PI solution had a concentration of 1 μm/mL. The *Arabidopsis* seedlings were soaked in PI solution for 1 and then used for micro-examination. mPS-PI staining was performed as described [47]. The seedlings of *Arabidopsis* were fixed in 50% methanol and 10% acetic acid for 30 min and vacuumized. Then, the seedlings were rinsed with water and transferred to 1% periodic acid for 40 min. Next, the seedlings were rinsed with water again and incubated in Schiff’s reagent (100 mM sodium metabisulphite and 0.15 N HCl) with propidium iodide (freshly added to a final concentration of 100 μg/mL) for 1 h until the plants were visibly stained. The plants were mounted in HCG for visualization.

### 4.4. Confocal Microscopy and Image Quantification

The *Arabidopsis* seedlings with GUS staining were viewed using a Nikon ECLIPSE Ni-U microscope connected to a Nikon DS-Ri2 digital camera (Nikon, Tokyo, Japan). Confocal laser scanning microscopy was performed on a Zeiss LSM880 (Zeiss, Oberkochen, Germany). Images were taken with a 40× water immersion objective.

## Figures and Tables

**Figure 1 plants-12-02365-f001:**
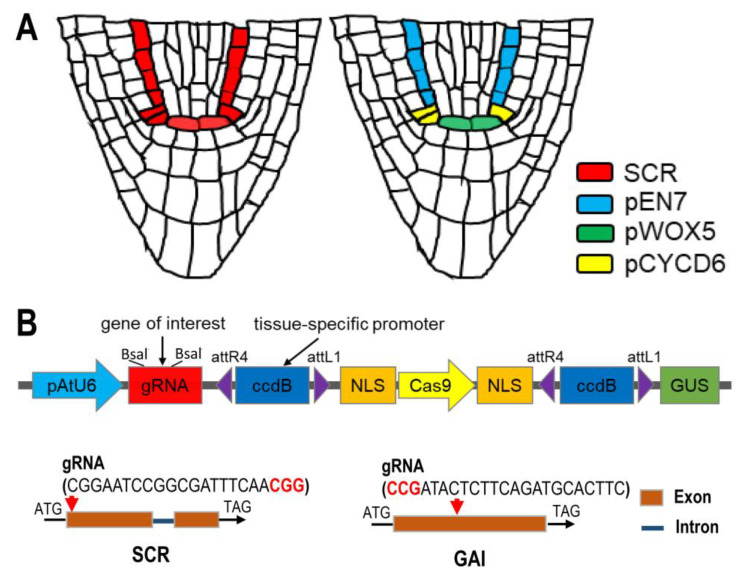
A CRISPR/Cas9 system for cell-type-specific gene disruption in *Arabidopsis thaliana*. (**A**) Schematic showing the expression pattern of SCR, EN7, WOX5, and CYCD6;1 in *Arabidopsis* roots. (**B**) Schematic of the cell-type-specific CRISPR/Cas9 vector. pAtU6-optimized U6 promoter; gRNA comprises two Bsal restriction sites allowing easy cloning of any gene-specific target sequence. All the schemes for vectors are in Appendix A. Schematic representation of the specific tissue knock-down of *SCR* and *GAI*. The orange box and blue line represent exons and introns, respectively. The red triangle indicates the physical location of the editing target. The editing sequence is shown above the target. The red bold fonts represent the PAM elements.

**Figure 2 plants-12-02365-f002:**
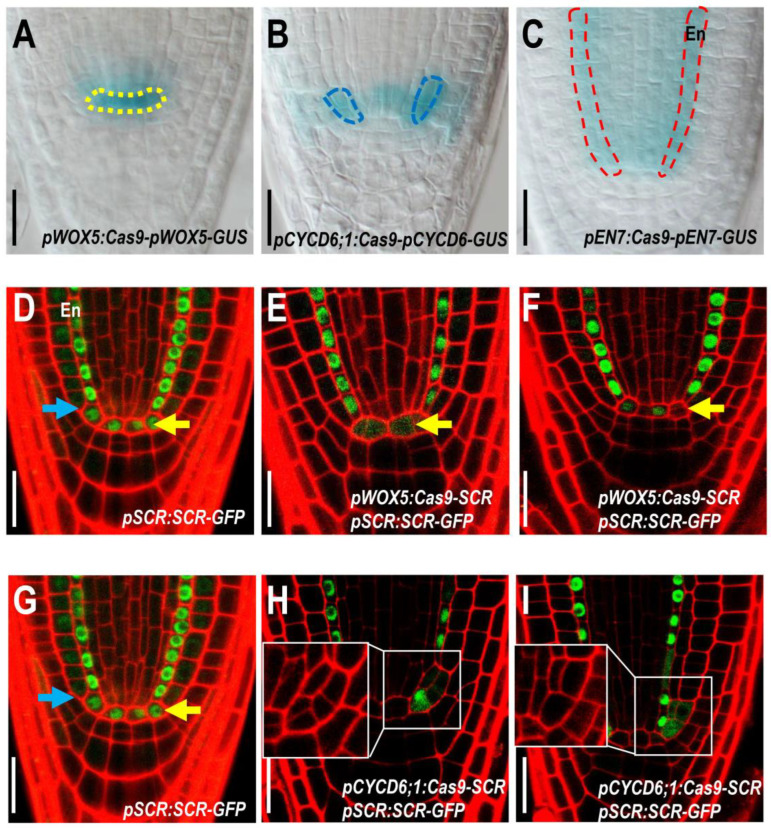
Specifically dissected SCR role in QC cells. (**A**–**C**) The specific expression driven by cell-type-specific promoters was approved by the QC-restricted GUS staining. The yellow box indicates the quiescent center (QC) cells, the blue box indicates the cortex/endodermis initial cells (CEIs), and the red box indicates the endodermis cells. “En” represents endodermis cells. (**D**–**F**) Expression of *pSCR:SCR-GFP* in WT (**D**) and *pWOX5:Cas9-SCR* (**E**,**F**) roots. The yellow arrows indicate QC cells and the blue arrows indicate cortex/endodermis initial cells (CEIs). (**G**–**I**) The expression observation of *pCYCD6;1:Cas9-SCR;pSCR:SCR-GFP*. Bars = 50 μm.

**Figure 3 plants-12-02365-f003:**
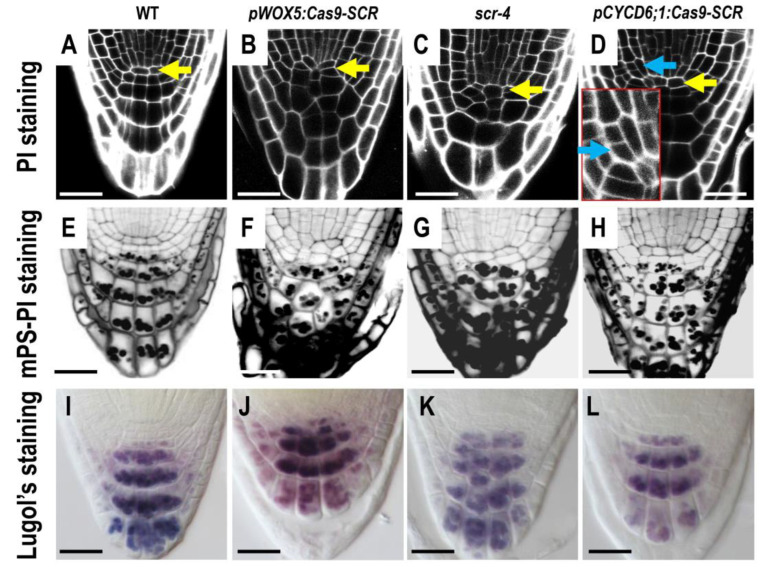
Identification of SCR function in different tissue cells. (**A**–**D**) Root phenotypes were analyzed by PI staining of WT (**A**), *pWOX5:Cas9-SCR* (**B**), *scr-4* (**C**), and *pCYCD6;1:Cas9-SCR* (**D**) roots. The yellow arrows indicate quiescent center (QC) cells and the blue arrows indicate cortex/endodermis initial cells (CEIs). The insets of D show the phenotype of CEIs. (**E**–**H**) mPS-PI staining of WT (**E**), *pWOX5:Cas9-SCR* (**F**), *scr-4* (**G**), and *pCYCD6;1:Cas9-SCR* (**H**) roots. (**I**–**L**) Lugol’s staining of WT (**I**), *pWOX5:Cas9-SCR* (**J**), *scr-4* (**K**), and *pCYCD6;1:Cas9-SCR* (**L**) roots. Bars = 50 μm.

**Figure 4 plants-12-02365-f004:**
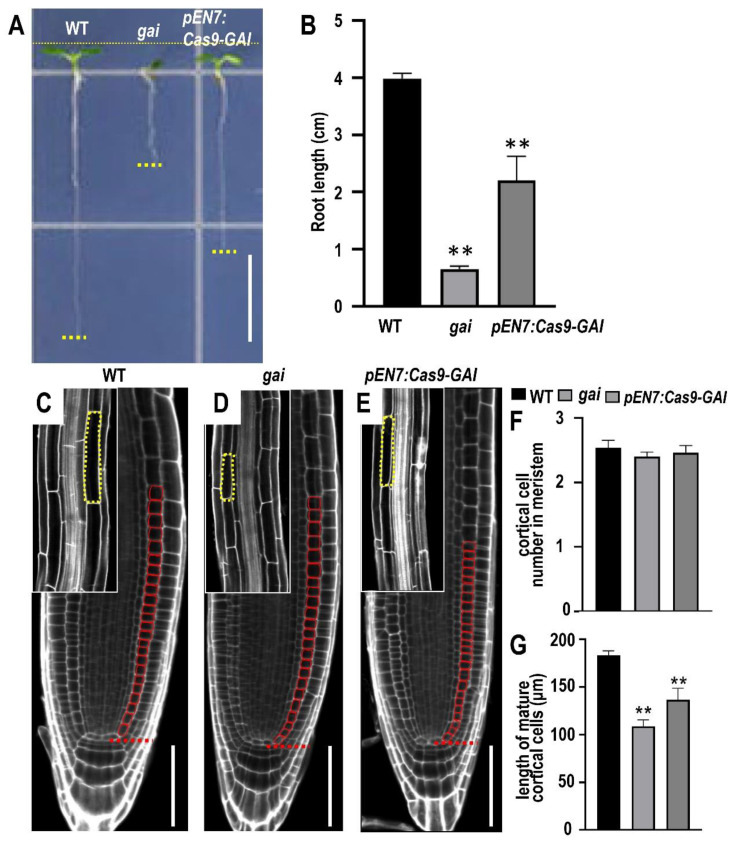
Specifically identified GAI role in endodermis cells. (**A**) Phenotypes of 10-day-old WT, *gai*, and *pEN7:Cas9-GAI* roots. The yellow dots indicate the end of root length. Bar = 1 cm. (**B**) Measurement of primary root length along WT, *gai*, and *pEN7:Cas9-GAI* roots. Error bars represent SD. (** *p <* 0.01, Student’s *t* test). (**C**–**E**) Propidium iodide-stained root meristem of WT, *gai*, and *pEN7:Cas9-GAI* roots. The red boxes indicate the cortical cell in the meristem and the yellow boxes indicate the cortical cell in the mature zone. Bars = 20 μm. (**F**,**G**) Measurements of meristem size by cortical cell number between the transition zone and the quiescent center. Measurements of cortical cell length within the mature zone of primary roots in 7 DAG seedlings. (** *p <* 0.01, Student’s *t* test).

**Table 1 plants-12-02365-t001:** Phenotypes of seedlings transformed with *pWOX5:Cas9-SCR*, *pCYCD6;1:Cas9-SCR*, and *pEN7:Cas9-GAI*.

Name	GUS	Root Number	Phenotypes	Editing Efficiency
*pWOX5* *:Cas9-SCR*	+	30	3	10%
*pCYCD6;1:Cas9-SCR*	+	30	5	16.1%
*pE* *N* *7:Cas9-GAI*	+	31	7	22.6%

## Data Availability

All supporting data are available from the corresponding author upon request.

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
