# Peer review of "CRISPR/Cas9 Based Cell-Type Specific Gene Knock-Out in Arabidopsis Roots"

_plants, 2023, doi:10.3390/plants12122365_

Round 1

Reviewer 1 Report (Previous Reviewer 1)

In the manuscript, Li et al. developed a CRISPR/Cas9-based platform to achieve gene knockout in a cell-type-specific manner for gene function study. They tested some cell-specific promoters such as WUSCHEL RELATED HOMEOBOX 5 (WOX5), CYCLIN D6;1 (CYCD6;1) and EN-DODERMIS7 (EN7) for Cas9 expression that allows target cell-specific gene expression. Furthermore, they claimed that tissue-specific promoter-driven Cas9 could make SCARE-CROW (SCR) and GIBBERELLIC ACID INSENSITIVE (GAI) knockout phenotype. Finally, they contended that the cell type-specific manipulation platform has excellent potential for studying spatiotemporal gene functions in multiallelic genetic backgrounds.

I appreciate the author’s efforts in conducting several experiments; however, I can point out some critical questions in this study.

Major comments

As a proof-of-concept, the authors used a well-known phenotypic marker gene to establish their system. However, to support the claim that “Specific knockout of SCR in QC leads to impaired QC function and SCN maintenance”, the author must provide the knockout gene data in the form of Sanger sequencing or next-generation sequencing (NGS) or both.

Often, cell-specific expression can be leaky. Therefore, evaluations of off-target effects are highly recommended.

Author Response

Thanks for your comment againbut we still thought that the Sanger/NGS sequencing is not technically feasible for our specific tissue knockout system. The specific reasons are as stated in the last reply letter. Firstly, it is extremely difficult to extract cells from the target tissue with GUS-tag for sequencing. Secondly, it is not possible to verify gene editing in the target tissue by sequencing, because for each individual cell in the target tissue, the editing of the knocked-out gene is an independent event, and the validation of the availability of tissue-specific knockout systems by means of whole-root sequencing has been denied in previous reports (Decaestecker et al., 2019).

We validated the usability of the system by observing the expression of the GUS-tag, and having observed a tissue-specific phenotype, the same approach to validating specific knockout system is widely recognized and applied (Feng Q et al., 2022; Wang et al., 2020; Nolan T M et al., 2023).

References:

Decaestecker, W., Buono, R. A., Pfeiffer, M., et al. CRISPR-TSKO: A Technique for Efficient Mutagenesis in Specific Cell Types, Tissues, or Organs in Arabidopsis. The Plant Cell, 2019, 31:2868–2887.

Feng Q, De Rycke R, Dagdas Y, et al. Autophagy promotes programmed cell death and corpse clearance in specific cell types of the Arabidopsis root cap. Current Biology, 2022, 32: 2110-2119.

Wang X, Ye L, Lyu M, et al. An inducible genome editing system for plants. Nature plants, 2020, 6: 766-772.

Nolan T M, Vukašinović N, Hsu C W, et al. Brassinosteroid gene regulatory networks at cellular resolution in the Arabidopsis root. Science, 2023, 379: eadf4721.

Reviewer 2 Report (Previous Reviewer 2)

The authors have satisfactorily addressed most of my concerns. I have minor corrections to be done listed below:

-Fig.1B and legend. Correct BasI to BsaI site. Please, also include PAM site sequences for both gRNAs. 

-GenBank No. for GAI gene must be corrected.

I highly recommend English proofreading by native speakers.

Author Response

-Fig.1B and legend. Correct BasI to BsaI site. Please, also include PAM site sequences for both gRNAs. 

Thank you for your comment. We have corrected Bsal site and added the PAM site sequences in this version.

-GenBank No. for GAI gene must be corrected.

We have corrected.

I highly recommend English proofreading by native speakers.

We have corrected the English expression.

Round 2

Reviewer 1 Report (Previous Reviewer 1)

I agree that harvesting target cells only to perform direct sequencing by Sanger sequencing is tricky. Although I believe that short reads, NGS such as MiniSeq can detect mutant alleles for the targets.

Authors suggested studies also provided genotype data to support their claim as follows.

Wang et al., Nature Plants, 2020 established inducible genome editing (IGE) system that enables efficient target gene knockouts in desired cell types in Arabidopsis. They also analysed the gene mutations using deep amplicon sequencing of the PLT2 genomic target site.

Decaestecker, The Plant Cell, 2019 also genotyped the segregating Arabidopsis plants to determine the frequency and type of knockout alleles generated. They reported indel frequencies in target sites.

Hence, I suggest harvesting the target tissue by a pool and performing the short reads, NGS to support their major claim of cell-specific gene knockout.   

Minor editing of English language required

Author Response

We thank the reviewer again for the comment! We respectfully disagree with that sequencing of PLT genes applies to the gene we examined in this study. Actually the difficulty in this paper is much more than the previous one, mainly due to the extremely restricted target cell numbers. The expression of PLT genes covers most of the root meristem, whereas WOX5 promoter or CYCD6;1 promoter are only active in a few cells in the root tip. That means PLT expressing cells may be thousands more than that of WOX5 or CYCD6;1 expressing cells. Therefore, it is technically difficult to do the sequencing in our system.

In addition, this reviewer also agrees that Sanger sequencing is unrealistic in this case, but suggests the use of very expensive next generation genome sequencing. Currently we have no funding for this type of sequencing, so we employed alternative way to test the editing. The alternative method we used here has also been used by a few studies before and they showed the validation of this marker-based method. In addition, we provided clear phenotypes in the target cells, demonstrating the usability of the system.

This manuscript is a resubmission of an earlier submission. The following is a list of the peer review reports and author responses from that submission.

Round 1

Reviewer 1 Report

The manuscript by Li et al. tested several cell-specific promoters such as WUSCHEL RELATED HOMEOBOX 5 (WOX5), CYCLIN D6;1 (CYCD6;1) and ENDO-14 DERMIS7 (EN7) for Cas9 gene expression to perform genome modification in a cell-specific manner. They fused Cas9 to GFP or GUS reporter and analyzed the cell-specific expression in Arabidopsis. As a proof of concept, authors tested their tissue-specific CRISPR/Cas9-based system to target developmental phenotypic marker genes SCARECROW (SCR) and GIBBERELLIC ACID INSENSITIVE (GAI).

The experimental methodology is correct; however, lacking enough data to claim, the presentation of the results, including discussions, needs revision. Please follow my comments. 

Major comments  

Authors took reported target genes to test their experiments; however, authors must include respective gene knockout data (Sanger/NGS), in the absence of such data this paper can not be accepted. 

pEn7:Cas9-GAI generated goi mutant showed intermediate phenotype compared with control goi mutant (Ubeda-Toma´s et al., 2009). Cell specific expression of Cas9 may lead to partial phenotype because of partial editing/ heterozygous genotype. Then authors must explain the limitations of their methods.

Minor

Please provide the abbreviation CRISPR/Cas9.

Line 55-57 Promoters can determine the gene expression quantitatively additionally temporally. However, germline-specific promoter-based expression of CRISPR reagents showed improved editing efficiency in segregating generations. I believe that "any" tissue-specific expression can not improve genome editing efficiency. 

Line 241, the reference is missing Yang L et al., 2023.

Reviewer 2 Report

Though the manuscript is generally good and interesting, the presentation of results and writing is obstructing the clear flow, and this degrades the overall quality of the paper. The manuscript needs extensive revision and overall structural refining.

In introduction, if we look at the study, it is ultimately about cell-specific expression of Cas9 in quiescent centers and endodermal cells. I suggest authors increase the introduction about the importance of these cells in plant physiology and the role of SCR and GAI genes in this regard. On the contrary, the first two paragraphs could be combined and shortened because there is no need to describe the value and mechanism of action of Cas9-mediated mutagenesis.

The majority of the Results actually represent poorly structured text that is mostly a mixture of introduction and discussion context. The Results section must be concise and only touch on your own data. With some exceptions, it should not contain any literature citations. Please, also divide this part of the manuscript into the particular subsections and give a more detailed interpretation of the obtained pictures and graphs. You could also consider combining Results and Discussion. Anyway, there should be a clear flow of what was done, what was obtained, and how you interpret the obtained findings in view of previously published literature. All the descriptive information needed to understand your results must be presented in the Introduction section. Additionally, in the case of SCR and GAI gene targeting, you should provide alternative mutation detection techniques like sequencing, fragment analysis of amplified PCR products, HRM-analysis, T7 Endonuclease cleavage of PCR products at sites of heteroduplex mismatch, etc.

I am sorry to say, but the Discussion is very poorly written. The authors must discuss the results by comparing their findings with those of similar previous studies and not just write another introduction to the Discussion. A conclusion paragraph also must deal with your own results instead of the main research questions.

In Materials and Methods, mention where you obtained the scr-4 and gai mutant plants. Provide all primer sequences. For genes, provide their GenBank Acc.No. There is no description of how gRNAs for GFP, SCR and GAI genes were chosen, and their sequences are also missing. Why is "Agro-bacterium" written with a hyphen? The strain's name is also spelled incorrectly.

In Fig.1B, provide schemes for all vectors you have used. For gRNAs provide their target DNA regions of interest and the selected exons.

In Fig. 4G error bars are missing.

Reviewer 3 Report

The study used cell specific promoter to drive the Cas9  and successfully precisely edit WOX5, CYCLIN D6;1 and ENDO-14 DERMIS7 genes. The system reported can be used for the type-specific manipulation, and the data generated in the study can help people to better understand the spatio and temporal functions of genes during plant root development. While the experiment was well-studied and results were significant, the literature review and discussion of the manuscript can be improved. Comments are listed as below:

Major:

1.        Citation of the most important reference (Decaestecker et al., 2019 Plant Cell) was missing (it was listed as reference, but not cited). The corresponding reference should be properly reviewed in Introduction and cited. The novelty of this study should be stated by comparing with the above reference in the Discussion.

2.        When use the cell-(tissue-)specific promoter to driving the CAS protein for gene editing, one major issue needs to be considered: if the efficiency of the specific editing lines. In Decaestecker et al., 2019, they achieved about 10% of the transgenic lines that are tissue-specifically edited.  The authors should present the ratio of the specific editing lines and discuss the result.  

Minor:

1.        Line63 Wrong citation

2.        Reference of Lockhart J 2019, and lots of others are incomplete;

3.        Reference is randomly ordered.

4.        Sentence in line 74-75 should not be there.

5.        Legend of Figure 1B, Schematic can not be used as a subject.

6.        Text (lines 113-118) and Legend of Figure 2F-I did not match with the label in the images.

7.        Revise citation, only need last name of the first author for the citation.